# Drone-Based 3D Synthetic Aperture Radar Imaging with Trajectory Optimization

**DOI:** 10.3390/s22186990

**Published:** 2022-09-15

**Authors:** Jedrzej Drozdowicz, Piotr Samczynski

**Affiliations:** Department of Electronics and Information Technology, Institute of Electronic Systems, Warsaw University of Technology, Nowowiejska 15/19, 00-665 Warsaw, Poland

**Keywords:** 3D, drone, imaging, optimization, radar, SAR, UAV

## Abstract

This paper presents a trajectory determination and optimization method of multirotors equipped with a single-channel radar to obtain 3D Synthetic Aperture Radar imaging. The result is a realistic trajectory that allows to obtain an imaging of the assumed quality in less time than using a multi-pass trajectory. The optimization criteria, in addition to the cross-range resolution, are the Peak Sidelobe Ratio (PSLR), Integrated Sidelobe Ratio (ISLR), and time of flight. The algorithm is based on a realistic motion model of the radar platform. This paper presents all the steps of the algorithm and provides simulation results that show its practical applicability. The advantage of the presented approach over the existing ones is indicated and further research directions are proposed.

## 1. Introduction

Synthetic Aperture Radar (SAR) technology has made significant progress from simple two-dimensional (2D) aerial imagery [1] to high-resolution three-dimensional (3D) drone imagery [2] in the last decades, and it is successfully becoming complementary to lidar as a three-dimensional (3D) imaging technique [3]. A commonly used three-dimensional (3D) Synthetic Aperture Radar (SAR) imaging method is Interferometric Synthetic Aperture Radar (InSAR), which is suitable for imaging the surface of the Earth. For urban area imaging, where Interferometric Synthetic Aperture Radar (InSAR) fails due to large height differences, tomographic Synthetic Aperture Radar (SAR) and Circular Synthetic Aperture Radar (CSAR) are commonly used. An important feature of these geometries is a very long, straight trajectory, and the high maneuverability of Unmanned Aerial Vehicles (UAVs), especially multirotors, is not fully utilized. The long flight time is a significant problem, especially for small UAVs with a limited battery capacity and significant radar power consumption.

The solution to these issues is to use a non-rectilinear trajectory of the radar platform, one that allows the imaging to be achieved with the assumed quality while being as short as possible. Because the multirotor can fly along any trajectory and its motion model is known, it is possible to optimize the trajectory and reduce its length. The effect is, first of all, energy savings and the possibility to perform imaging without a break for battery charging but also a reduction in the amount of recorded data, which speeds up the processing.

This paper presents a method of determining the non-rectilinear trajectory of a radar platform to obtain three-dimensional (3D) images of specific areas, with assumed quality. The main contribution of this article is the method of determining the trajectory, taking into account the radar platform’s (multirotor) motion model and the method of optimizing this trajectory.

The remainder of this section introduces the existing three-dimensional (3D) Synthetic Aperture Radar (SAR) imaging methods and covers the state of the art. The next section describes in detail the three-dimensional (3D) Synthetic Aperture Radar (SAR) imaging method, using a non-rectilinear trajectory of the radar platform along with the method of its derivation, trajectory determination, and optimization. The next section presents the simulation results of the presented method, developed using a realistic model of the motion of the radar platform. The last section is devoted to a discussion of the results and a summary.

### 1.1. Synthetic Aperture Radar Signal Model

Synthetic Aperture Radar (SAR) is a technique for obtaining radar images using a radar sensor placed on a moving radar platform. It should be stressed that the movement of the platform itself is not necessary to obtain a Synthetic Aperture Radar (SAR) image, but only the fact that the radar sensor is placed at different positions [4]. A synthetic aperture comprising those positions produces a Point Spread Function (PSF) in the form of [5]: (1)$$\begin{array}{c} E\left(p\right)=\Sigma_{n=1}^{n=N}\exp\left(j2\pi \frac{r_{n}}{\lambda }\right)\mathrm{sinc}\left(\frac{r_{n}}{\frac{c}{2B}}\right), \end{array}$$
where p denotes the point at which the PSF is calculated; $r_{n}=|p-p_{an}\left|-\right|p_{t}-p_{an}|$ is the difference between the distance from the point p to the n−th antenna position $p_{an}$ and the distance from the point target $p_{t}$ to the n−th antenna position $p_{an}$; λ is the wavelength; *c* is the speed of light; and *B* is the signal bandwidth. The most common waveform type for the Synthetic Aperture Radar (SAR) is Frequency Modulated Continuous Wave (FMCW) [6] in the form of
(2)$$\begin{array}{c} x\left(t\right)=A\mathrm{rect}\left(\frac{t}{T}\right)\cos\left(2\pi f_{0}t+\pi \alpha t^{2}\right), \end{array}$$
where $f_{c}$ is the carrier (center) frequency, $\alpha =\frac{B}{T}$ is the LFM slope, *B* is the bandwidth, *T* is the pulse duration, and *A* is the signal amplitude. Other types of radar are also used, such as noise [7] and passive [8]. This paper is not specifically related to any of these methods and assumes that all conditions for the validity of (Equation 1) are met and that the resolution of the Synthetic Aperture Radar (SAR) image depends solely on the trajectory of the radar carrier, wavelength, and signal bandwidth. Such conditions include, e.g., a time-bandwidth product higher than approximately 100 (BT≳100) for FMCW [4].

### 1.2. Existing 3D SAR Imaging Methods

A one-dimensional (1D) synthetic aperture (a straight line) produces two-dimensional (2D) Synthetic Aperture Radar (SAR) image and a two-dimensional (2D) synthetic aperture can produce a three-dimensional (3D) Synthetic Aperture Radar (SAR) image. The most popular approach to three-dimensional (3D) Synthetic Aperture Radar (SAR) imaging is Interferometric Synthetic Aperture Radar (InSAR) [6], which requires at least two receiving antennas or at least two consecutive passes of the radar carrier, as presented in Figure 1.

The vertical resolution of Interferometric Synthetic Aperture Radar (InSAR) is [6]: (3)$$\delta_{V}=\frac{\lambda r_{0}}{2L_{V}},$$
where *r_0_* is the distance to the scene and $L_{V}$ is the distance between the antennas (baseline). For the two-antenna or two-pass case, Equation (Equation 3) also defines the maximum unambiguous height of the scene [6]: (4)$${h_{max}}_{V}=\frac{\lambda r_{0}}{2d_{V}},$$
where $d_{V}$ is the distance between the antennas (note that in this case $d_{V}=L_{V}$). This can be seen in Figure 2 where the mainlobe is duplicated with a raster equal to its width, which means that the maximum unambiguous height is equal to the resolution (unless noted otherwise, simulations in this paper use the following radar parameters: carrier frequency fc=3 GHz, bandwidth B=300 MHz, and range $r_{0}=100$ m; simulations assume that there is no signal noise). To overcome height ambiguity, phase unwrapping can be used; however, in urban areas where height differences are significant, such unwrapping does not produce reliable results. In such cases, a compromise between resolution and height ambiguity has to be made: a large distance between antennas/passes (baseline) ensures high resolution, while high unambiguous height requires a short baseline. To avoid the need for the aforementioned compromise, CSAR or Multi-Baseline Synthetic Aperture Radar (MBSAR) can be used [9,10]. An example of such a trajectory is presented in Figure 3.

In this case, the distance between passes ($d_{V}$) is not equal to the vertical length of the synthetic aperture ($L_{V}$). An example of PSF is presented in Figure 4.

It can be seen that increasing the vertical length of the synthetic aperture ($L_{V}$) by adding more baselines increases the resolution without increasing the height ambiguity—the distance between the repeated lobes ${h_{max}}_{V}$ is the same in Figure 2 and Figure 4 (in the first case, ${h_{max}}_{V}=\delta_{V}$—see (Equation 3) and (Equation 4)), but the size of the lobe $\delta_{V}$ is smaller in the second case. The main drawback of these particular methods is the unnecessarily long scanning time and the large amount of data.

Another solution is the use of non-rectilinear trajectories. The literature available on that topic is scarce; however, two sources provide a decent basis for further development. The research presented in [11] focuses on bistatic three-dimensional (3D) Synthetic Aperture Radar (SAR) imaging and a UAV is considered as a RAdio Detection And Ranging (radar) carrier. In [11], both parametric and nonparametric imaging are considered, and trajectory non-rectilinearity is introduced by adding curvature. This approach is an important step to establish the relationship between the three-dimensional (3D) image quality and the receiver flight path. Another study [12] focuses on a sinusoidal trajectory (see Figure 5) and presents numerous simulation results of imaging simple scenes with isotropic reflecting points for trajectories that are a single period of a sinusoidal function, a segment of a circle, and a combination (sum) of sinusoidal and circular trajectories. Although the article often emphasizes the strong dependence of the characteristics on the aspect angle, the simulations were carried out only for isotropic points. The problem of the possibility of implementing a given trajectory with a radar carrier was raised, but the only conclusion presented was the need to minimize the height gradient, indicating that a fixed-wing aircraft was probably considered a radar carrier.

Both approaches mentioned above do not fully exploit the potential of a multirotor that can fly along trajectories much more complex than a curved line [13]. What is more, the practical application of such trajectories can be difficult, because they are determined without taking into account the platform motion model. In the case of fixed wings, following the curve accurately requires solving a series of equations, while in the case of multirotors, it requires determining many waypoints.

### 1.3. Imaging Algorithms

To obtain a Synthetic Aperture Radar (SAR) image from reflected echoes (or the beat signal), an imaging algorithm must be applied. For a typical two-dimensional (2D) Synthetic Aperture Radar (SAR), a matched filter method can be applied that is often described as the fastest method producing high-resolution images [14]. An Interferometric Synthetic Aperture Radar (InSAR) image is usually obtained by determining the phase difference between two images from two antennas or passes, and a similar approach can be applied to CSAR and MBSAR. In the case of a non-rectilinear trajectory, this method is not applicable, and other methods have to be used. The simplest but most computationally demanding method is the Back Projection Algorithm (BPA), which relies on the direct application of (Equation 1) [8]. An improvement in terms of computational complexity is provided by the Polar Format Algorithm (PFA) [15]. Both the BPA and PFA allow for an arbitrary image pixel size, where in the case of the matched filter method, the pixel size is determined by the radar parameters [6].

## 2. Method

The trajectories described in the previous section are simple, but because they are not sufficiently parameterizable, they are not suitable for advanced optimization. The trajectories presented so far are the result of changing the position of the radar platform over time. However, each trajectory can be considered in isolation from time, as a set of points in space, i.e., a synthetic aperture [4]. Following this line of reasoning, each synthetic aperture is an example of sampling a certain surface (synthetic aperture surface). The shape (outline) of the surface corresponds to the mainlobe of PSF, while the distribution of sampling points corresponds to the sidelobes (or grating lobes in the case of even sampling). An example is presented in Figure 6.

Evident grating lobes can be seen. If uniform sampling is replaced by random sampling (Figure 7), the grating lobes become smeared. In general, a sufficiently densely sampled aperture of dimension $L_{H}$ by $L_{V}$ provides $\delta_{H}$ by $\delta_{V}$ resolution when: (5)$$\begin{array}{cc} L_{H}= & \frac{r_{0}\lambda }{2\delta_{H}}, \end{array}$$(6)$$\begin{array}{cc} L_{V}= & \frac{r_{0}\lambda }{2\delta_{V}}, \end{array}$$
where $\delta_{H}$ and $\delta_{V}$ are the horizontal and vertical resolutions, respectively, and *r*_0_ is the distance from the aperture to the target. Note that both multiple rectilinear flights at different altitudes (Figure 4) and a flight along some non-rectilinear trajectories are examples of aperture sampling (Figure 8).

This provides an introduction to the method described in this paper. It assumes the determination of a radar platform trajectory that samples the aperture surface in order to achieve the quality requirements. The block diagram of the algorithm is shown in Figure 9.

The algorithm starts with the determination of the synthetic aperture surface, and then the waypoints are placed on it through which the trajectory is carried out according to the motion model of the platform. The final step is optimization to improve the quality parameters and reduce the flight time. The subsequent stages are described in the following subsections.

### 2.1. Synthetic Aperture Surface Determination

The synthetic aperture surface outline is characterized by the distance from the imaged object and the span. The distance from the object can be taken arbitrarily, with the requirement that the entire object of size $D_{H}$ by $D_{V}$ fits within the antenna beam:(7)$$\begin{array}{cc} r_{0} & >\frac{D_{V}}{2}\tan\left(\frac{\theta_{V}}{2}\right), \end{array}$$(8)$$\begin{array}{cc} r_{0} & >\frac{D_{H}}{2}\tan\left(\frac{\theta_{H}}{2}\right), \end{array}$$
where $D_{V}$ and $D_{H}$ are vertical and horizontal dimensions of Region Of Interest (ROI), and $\theta_{V}$ and $\theta_{H}$ are vertical and horizontal antenna beamwidths. Far-field conditions are assumed to be met: (9)$$\begin{array}{cc} r_{0} & \gg D, \end{array}$$(10)$$\begin{array}{cc} r_{0} & \gg \lambda , \end{array}$$
where *r*_0_ is the distance to the object and D is the largest ROI dimension. The aperture span is determined according to the Formulas (Equation 6) and (Equation 5).

### 2.2. Trajectory Determination

From a practical point of view, it is important that the trajectory is defined unambiguously by a minimum number of parameters, as the number of parameters determines the number of dimensions of the function to be optimized. If the motion model of the radar platform is known, and the influence of atmospheric conditions (wind) on the actual trajectory can be neglected, the trajectory is unambiguously determined by the location of waypoints. Waypoints are the navigation points used by the radar platforms’ autopilot to create the trajectory. Waypoints are visited by the platform in predefined order and a waypoint is considered visited when the distance between the platform and the waypoint is lower than specified acceptance radius. In this paper, the number of waypoints is indicated by *N* and they are distributed along the horizontal dimension of the aperture plane, placed alternately on the lower and upper sides of it, visited by the platform according to a zigzag pattern (see Figure 8), where the positions of the initial ($P_{W1}$) and final ($P_{WN}$) points are given by: (11)$$\begin{array}{cc} P_{W1} & =\left(\begin{array}{c} -r_{0}\\ -\frac{L_{H}}{2}\\ -\frac{L_{V}}{2} \end{array}\right), \end{array}$$(12)$$\begin{array}{cc} P_{WN} & =\left\{\begin{array}{cc} \left(\begin{array}{c} -r_{0}\\ \frac{L_{H}}{2}\\ \frac{L_{V}}{2} \end{array}\right) & \mathrm{if}\;2|N\\ \left(\begin{array}{c} -r_{0}\\ \frac{L_{H}}{2}\\ -\frac{L_{V}}{2} \end{array}\right) & \mathrm{if}\;2∤N, \end{array}\right. \end{array}$$
where *r* is the distance to the object, $L_{H}$ and $L_{V}$ are the horizontal and vertical aperture dimensions, respectively, and 2∣N means *N is divisible by 2* and 2∤N means *N is not divisible by 2*. The positions of the rest of the points: $$\begin{array}{cc} P_{Wn} & =\left(\begin{array}{c} P_{Wnx}\\ P_{Wny}\\ P_{Wnz} \end{array}\right) \end{array}$$
are determined by: (13)$$\begin{array}{cc} P_{Wnx} & =-R \end{array}$$(14)$$\begin{array}{cc} P(P_{Wny}<\gamma ) & =\left\{\begin{array}{cc} 0 & \mathrm{for}\;\gamma \leq -\frac{L_{H}}{2}\\ \frac{\gamma }{L_{H}}+\frac{1}{2} & \mathrm{for}\;-\frac{L_{H}}{2}<\gamma \leq \frac{L_{H}}{2}\\ 1 & \mathrm{for}\;\gamma >\frac{L_{H}}{2} \end{array}\right. \end{array}$$(15)$$\begin{array}{cc} P_{Wiy}\leq P_{Wjy} & \forall i<j \end{array}$$(16)$$\begin{array}{cc} P_{Wnz} & =\left\{\begin{array}{cc} -\frac{L_{V}}{2} & \mathrm{for}\;2|n\\ \frac{L_{V}}{2} & \mathrm{for}\;2∤n, \end{array}\right. \end{array}$$
where operator P(·) denotes probability and γ is an independent variable used only for definition. The horizontal position of the points is random (non-uniform). If it were uniform, one would obtain a trajectory close to the sinusoidal one, as described in [12], which is characterized by very prominent sidelobes (see Figure 5). To counteract this, the points are randomly distributed, which reduces the prominence of the strongest sidelobe, at the expense of introducing multiple weaker sidelobes as seen in Figure 8.

### 2.3. Waypoints Following

In this paper, the following waypoint-following model is adopted: only one consecutive waypoint is active at each moment of platform movement, the velocity and acceleration are chosen so that the platform can stop when reaching a waypoint. An acceptance radius is defined for each waypoint; if the distance of the platform from the waypoint is smaller than the acceptance radius, the next waypoint becomes active. A demonstration of the motion model is presented in Figure 10.

The trajectory calculation process has the following steps for each discrete time moment *t*:Check if the current waypoint has been reached:
(17)$$\begin{array}{c} |P_{P}\left(t\right)-P_{Wn}|\leq r_{a}\Rightarrow n:=n+1 \end{array}$$
(18)$$\begin{array}{c} n>N\Rightarrow \mathrm{trajectory}\;\mathrm{complete}, \end{array}$$
where $P_{P}\left(t\right)$ is the position of the platform at time *t*, $P_{Wn}$ is the *n*-th waypoint, $r_{a}$ is the radius of the acceptance circle, and *N* is the total number of waypoints.Determine the directional vector of the trajectory that points toward current waypoint:
(19)$$\begin{array}{c} d=\frac{P_{P}\left(t\right)-P_{Wn}}{\left\|P_{P}\left(t\right)-P_{Wn}\right\|}. \end{array}$$Determine maximum speed in that direction:
(20)$$\begin{array}{c} v_{m}=\sqrt{2a_{max}\left(-d\right)s_{b}\left(d\right)}, \end{array}$$
where $a_{max}\left(\cdot \right)$ is the maximum acceleration in the specified direction (motion model limit) and $s_{b}\left(\cdot \right)$ is the braking distance-distance required for the radar platform to stop when moving with the maximum speed:
(21)$$\begin{array}{c} s_{b}=\frac{1}{2}\frac{v_{max}^{2}\left(d\right)}{a_{max}\left(-d\right)}, \end{array}$$
where $v_{max}\left(\cdot \right)$ is the maximum velocity in the specified direction (motion model limit).Determine target velocity:
(22)$$\begin{array}{c} v_{t}=\left\{\begin{array}{cc} \sqrt{2a_{max}\left(-d\right)\left(P_{P}\left(t\right)-P_{Wn}\right)}d & \mathrm{if}\;P_{P}\left(t\right)-P_{Wn}\leq r_{a}\\ \min\left(v_{m},v_{max}\left(d\right)\right)d & \mathrm{otherwise}. \end{array}\right. \end{array}$$Determine the difference between the target velocity and the current velocity:
(23)$$\begin{array}{c} \Delta v=v_{t}-v\left(t\right), \end{array}$$
where v(t) is the radar platform velocity at time *t* (current velocity).Determine the target acceleration:
(24)$$\begin{array}{c} a_{t}=\frac{\Delta v}{\left\|\Delta v\right\|}a_{max}\left(\frac{\Delta v}{\left\|\Delta v\right\|}\right) \end{array}$$
and clip if it is too high:
(25)$$\begin{array}{c} a\left(t+\Delta t\right)=\left\{\begin{array}{cc} \frac{\left\|\Delta v\right\|}{\left\|a_{t}\right\|}a_{t} & \mathrm{if}\;\left\|a_{t}\Delta t\right\|>\left\|\Delta v\right\|\\ a_{t} & \mathrm{otherwise}, \end{array}\right. \end{array}$$
where Δt is the calculation time step.Finally, determine updated velocity and position:
(26)$$\begin{array}{cc} v(t+\Delta t) & =v(t)+a(t+\Delta t)\Delta t \end{array}$$
(27)$$\begin{array}{cc} P_{P}\left(t+\Delta t\right) & =P_{P}\left(t\right)+v\left(t+\Delta t\right)\Delta t. \end{array}$$

### 2.4. Quality Assessment

With the trajectory determined, its PSF can be calculated so that quality metrics can be assigned, allowing optimization. The PSF is determined according to Formula (Equation 1) in the x=0 plane, in the range: (28)$$\begin{array}{cc} y & \in \left(-\frac{D_{H}}{2},\frac{D_{H}}{2}\right), \end{array}$$(29)$$\begin{array}{cc} z & \in \left(-\frac{D_{V}}{2},\frac{D_{V}}{2}\right), \end{array}$$
where $D_{H}$ and $D_{V}$ are the horizontal and vertical dimensions of ROI, respectively. PSF can be determined with any resolution; the finer, the better. In this study, it was empirically found that an acceptable compromise between speed and accuracy of the calculation is adopting a pixel size at least ten times smaller than the expected smallest resolution.

### 2.5. Mainlobe Extraction

Mainlobe extraction is required to determine all quality parameters. For this purpose, the standard recursive flood-fill algorithm was used, which is as follows.
Mark the center pixel as belonging to the mainlobe;Mark each neighboring pixel with a smaller value as belonging to the mainlobe;Start from step 2 for each marked neighboring pixel.
With the mainlobe extracted, quality parameters can be estimated.

### 2.6. Resolution Estimation

Because the mainlobe shape deviates from a rectangle when SA is sampled unevenly, it is necessary to develop an equivalent measure of resolution. For the purpose of this research, it is defined as a rectangle whose outline is most similar to that of the mainlobe. Vertical and horizontal resolutions are determined by $\tilde{\delta_{V}}\left(\alpha_{0}\right)$ and $\tilde{\delta_{H}}\left(\alpha_{0}\right)$, respectively. $\alpha_{0}$ denotes the angle at which the condition of the best similarity is met: (30)$$\begin{array}{c} \alpha_{0}=\underset{\alpha \in (0,\frac{\pi }{2})}{\arg\min}\left(\arctan(\frac{\tilde{\delta_{V}}\left(\alpha \right)}{\tilde{\delta_{H}}\left(\alpha \right)})-\alpha \right), \end{array}$$
where $\tilde{\delta_{V}}\left(\alpha \right)$ and $\tilde{\delta_{H}}\left(\alpha \right)$ are the vertical and horizontal dimensions of assumed rectangle, calculated as the average of *z* and *y* coordinates of the points belonging to the vertical and horizontal mainlobe outline, respectively: (31)$$\begin{array}{c} \tilde{\delta_{V}}\left(\alpha \right)=\bar{|{{p_{o}}_{V}}_{z}|}, \end{array}$$(32)$$\begin{array}{c} \tilde{\delta_{H}}\left(\alpha \right)=\bar{|{{p_{o}}_{H}}_{y}|}. \end{array}$$

Vertical and horizontal outlines ${p_{o}}_{V}$ and ${p_{o}}_{H}$ for given α are the sets of points defined as “top” and “right” sides of mainlobe outline: (33)$$\begin{array}{c} {p_{o}}_{i}\in {p_{o}}_{V}\Leftrightarrow \arctan|\frac{{p_{o}}_{iz}}{{p_{o}}_{iy}}|>\alpha , \end{array}$$(34)$$\begin{array}{c} {p_{o}}_{i}\in {p_{o}}_{H}\Leftrightarrow \arctan|\frac{{p_{o}}_{iz}}{{p_{o}}_{iy}}|<\alpha . \end{array}$$

The outline of the mainlobe is obtained by determining those PSF pixels $p_{i}$ that do not belong to the mainlobe (ML) but are adjacent to at least one pixel belonging to the mainlobe: (35)$$\begin{array}{c} p_{i}\in p_{o}\Leftrightarrow p_{i}\notin p_{ML}\wedge \exists p_{j}\in p_{ML}:p_{j}=p_{i}+\delta ,\delta \in \left(1,0),(0,1),(-1,0),(0,-1\right), \end{array}$$
where $p_{o}$ is the set of pixels belonging to the outline of the mainlobe, $p_{ML}$ is the set of pixels belonging to the mainlobe. This provides a measure that is equivalent to the resolution that would be achieved with a rectangular aperture, which PSF is the most similar to the PSF under consideration.

### 2.7. PSLR and ISLR Estimation

Peak Sidelobe Ratio (PSLR) describes how prominent the strongest sidelobe is and is determined by Formula [16]: (36)$$PSLR=20\log_{10}\left(\frac{\max_{p\in p_{SL}}\left[E\left(p\right)\right]}{\max_{p\in p_{ML}}\left[E\left(p\right)\right]}\right),$$
whereas Integrated Sidelobe Ratio (ISLR) describes the overall level of the sidelobes and is determined by Formula [16]: (37)$$ISLR=20\log_{10}\left(\frac{\sum_{p\in p_{SL}}\left[E\left(p\right)\right]}{\sum_{p\in p_{ML}}\left[E\left(p\right)\right]}\right),$$
where E(p) is the PSF at the point p $p\in p_{SL}$ denotes points in the sidelobe and $p\in p_{ML}$ denotes points in the mainlobe.

### 2.8. Trajectory Cost

Once the quality parameters are known, the value of the cost function can be determined. The value of the cost function for a given trajectory is the sum of the products of the transfer function values for each parameter and their weights.
(38)$$\begin{array}{c} c=w_{\delta_{H}}c_{\delta_{H}}+w_{\delta_{V}}c_{\delta_{V}}+w_{PSLR}c_{PSLR}+w_{ISLR}c_{ISLR}+w_{t}c_{t}, \end{array}$$
where:$c_{\delta_{H}}$ is the cost related to the horizontal resolution;$c_{\delta_{V}}$ is the cost related to the vertical resolution;$c_{PSLR}$ is the cost related to the Peak Sidelobe Ratio (PSLR);$c_{ISLR}$ is the cost related to the Integrated Sidelobe Ratio (ISLR);$c_{t}$ is the cost related to the trajectory length (flight time, energy consumption).
and $w_{\delta_{H}}$, $w_{\delta_{V}}$, $w_{PSLR}$, $w_{ISLR}$, $w_{t}$ are the corresponding weighting factors. This set of optimization parameters (resolution, Peak Sidelobe Ratio (PSLR), Integrated Sidelobe Ratio (ISLR), flight time) was arbitrarily chosen by the authors because they are meaningful and are directly linked with the trajectory shape. Other important image-quality parameters, such as Imange Contrast (IC) and Image Entropy (IE), are not as affected by the shape of the trajectory.

The weighting factors and fractional cost functions must be defined in a way that ensures the proper optimization goal, that the optimizers main goal is to ensure the proper imaging quality and that the others costs do not prevail.

### 2.9. Transfer Functions

Trajectory functions are used to assign each trajectory parameter a cost value that is comparable in magnitude so that the cost of the trajectory can be determined. Two types of transfer functions are used in the model presented in this paper, a range-type transfer function and a step-type transfer function.

#### 2.9.1. Range-Type Transfer Function

Range-type transfer function is defined as
(39)$$\begin{array}{c} c=\left\{\begin{array}{cc} 0 & \mathrm{if}\;v<v_{L}\\ \frac{v-v_{L}}{v_{U}-v_{L}} & \mathrm{if}\;v_{L}\leq v\leq v_{U}\\ 1000\frac{v-v_{U}}{v_{U}-v_{L}}+1 & \mathrm{if}\;v_{U}<v, \end{array}\right. \end{array}$$
where c is cost, v is value (quality parameter), $v_{L}$ is the lower limit of the acceptable range of values, and $v_{U}$ is the upper limit. The purpose of using this function is to ensure that the cost varies linearly from 0 to 1 in the range from $v_{L}$ to $v_{U}$. Below $v_{L}$, the cost is 0, so that the optimizer does not attempt to reduce the parameter below this value, while above $v_{U}$, the cost increases linearly with a factor of 1000 (chosen arbitrarily to be large, but not large enough to hinder the optimizer), so that the optimizer will devote considerable effort to reducing the parameter to an acceptable level. This function should be applied to parameters for which an acceptable range of values can be specified, namely resolution, Peak Sidelobe Ratio (PSLR), and Integrated Sidelobe Ratio (ISLR). An example graph of the range transfer function is shown in Figure 11.

#### 2.9.2. Step-Type Transfer Function

The step-type transfer function is used to determine the cost of the flight time and is defined as
(40)$$c=\left\{\begin{array}{cc} \left\lfloor \frac{v}{v_{U}}\right\rfloor \left(1+c_{s}\right)+\frac{v\;\mathrm{mod}\;v_{U}}{v_{U}} & \mathrm{if}v\;\mathrm{mod}v_{U}\leq v_{U}-\frac{c_{S}v_{U}}{1000}\\ \left\lfloor \frac{v}{v_{U}}\right\rfloor \left(1+c_{s}\right)+1000\frac{v\;\mathrm{mod}\;v_{U}}{v_{U}}+1+c_{s}-1 & \text{otherwise} \end{array}\right.$$
where *c* is cost, *v* is value (flight time), where *v_U_* is the platform operating time provided by a full battery (or other energy source) and *c_S_* is a parameter that specifies the time required to return to base and charge the battery relative to the cost of consuming the entire battery. It was introduced in order to allow the analysis to take into account the situation when it is not possible to perform the entire measurement in one go and should be set to zero if this should not be taken into account. The step-type transfer function is presented in Figure 12.

### 2.10. Optimization

Trajectory optimization involves minimizing a multidimensional, nonlinear, non-convex cost function whose argument is the waypoint location: (41)$$\begin{array}{c} P_{WOpt}=\underset{P}{arg min}\;c\left(P\right), \end{array}$$
where $P_{WOpt}$ is the set of optimized waypoints and P is the superset of the set of waypoints covered by the optimization procedure. There are numerous algorithms for optimizing convex problems [17]. Some of them require the function to be differentiable, and others do not. A much more serious issue is non-convex optimization. There are various approaches, the most popular of which is to divide the function area into subareas, using different starting points, and applying convex optimization. Subarea partitioning can be implemented in a way that ensures that a global extremum is reached, but it requires a deep analysis of the function. In this paper, the approach of using different starting points is used.

The Nelder–Mead algorithm [18], available as part of the MATLAB package [19], was used to optimize (minimize) the cost function. This algorithm can be used successfully to minimize non-differentiable functions, as it does not require the determination of a gradient. It uses a simplex, a 2N-dimensional shape in the input data space (two coordinates of N waypoints form a 2N-dimensional space), at whose vertices it determines the values of the minimized function. Subsequent steps of the algorithm move the simplex or change its shape to reach a local minimum. Simplified steps of the algorithm minimizing function c(P) are shown below:Initial simplex: create a simplex consisting of 2N+1 points around a starting point, including the starting point.Sort the points so that $c\left(P_{1}\right)\leq c\left(P_{2}\right)\leq \ldots \leq c\left(P_{2N+1}\right)$.Determine the centroid $P_{0}$ of the simplex, excluding the worst point $P_{2N+1}$: $P_{0}=A\left(P_{1}\ldots P_{2N}\right)$, where A(·) denotes mean.Reflection: determine the reflected point $P_{r}=P_{0}+k_{\alpha }\left(P_{0}-P_{2N+1}\right)$, where $k_{\alpha }$ is the reflection coefficient. If $P_{r}$ is the best point $c\left(P_{r}\right)<c\left(P_{1}\right)$, go to step 5. If it is the worst point $c\left(P_{r}\right)>c\left(P_{2N+1}\right)$, go to step 6. If it is the second worst $c\left(P_{2N}\right)<c\left(P_{r}\right)<c\left(P_{2N+1}\right)$, go to step 7. Otherwise, add it to the simplex in place of $P_{2N+1}$ and go back to step 1.Expansion: determine the expanded point: $P_{e}=P_{0}+k_{\gamma }\left(P_{r}-P_{0}\right)$, where $k_{\gamma }$ is the expansion coefficient. If it is better than reflected, add it in place of $P_{2N+1}$. Otherwise, add reflected. Return to step 1.Contract inside: determine the contracted point $P_{c}=P_{0}+k_{\rho }\left(P_{2N+1}-P_{0}\right)$, where $k_{\rho }$ is the contraction coefficient, and if it is not the worst, put it in place of the worst and return to step 1. Otherwise, go to step 8.Contract outside: determine the contracted point $P_{c}=P_{0}+k_{\rho }\left(P_{r}-P_{0}\right)$, where $k_{\rho }$ is the contraction coefficient, and if it is better than reflected, replace the worst with it and return to step 1. Otherwise, go to step 8.Shrink: reduce the size of the simplex: $\forall i>1:P_{i}=P_{1}+k_{\sigma }\left(P_{i}-P_{1}\right)$, where $k_{\sigma }$ is shrink coefficient.

The algorithm stops on two conditions: The iteration limit has been reached, or both the simplex size and the difference between the function values are below certain thresholds: $\max(|P_{i}-P_{j}\left|\right)<T_{P}$ and $|c\left(P_{2N+1}\right)-c\left(P_{1}\right)|<T_{c}$, where $T_{P}$ is the argument step threshold and $T_{c}$ is the function step threshold. The values used in this experiment are as follows: the iteration limit is 3000, $T_{P}=T_{c}=0.0001$, $k_{\alpha }=1$, $k_{\gamma }=2$, $k_{\rho }=0.5$, $k_{\sigma }=0.5$.

## 3. Results

### 3.1. Performance Analysis

The algorithm was tested for a test case with the radar and geometry parameters summarized in Table 1. A C–band radar with a low bandwidth was chosen as an example of a low-cost, lightweight sensor. However, while higher-band, low-cost, lightweight radars certainly exist, they require very high precision (sub-cm) GNSS platforms, which are neither low cost nor lightweight.

The optimization requirements were determined as follows: First, an MBSAR reference aperture was prepared to ensure that there are no grating lobes, that is, the vertical sampling distance (the distance between consecutive passes) is
(42)$$d_{V}=\frac{{\lambda r}_{0}}{2D_{V}},$$
where *r_0_* is the distance from the synthetic aperture to the object and $D_{V}$ is the vertical size of the object. For this aperture, the quality parameters, denoted further by the subscript ref, were determined. Then, the initial aperture was generated, and for it, the quality parameters, denoted further by the subscript ini, were also determined. Based on these results, the optimization parameters are defined and collected in Table 2.

The next step is to perform the optimization, where the cost is determined according to (Equation 38), and the variables are the y and z coordinates of the waypoints. The optimization was performed in MATLAB using fminsearch [19], that is, using the derivative-free Nelder–Mead simplex algorithm [18]. The optimization was repeated ten times for different starting points. The results are summarized in Table 3.

A graph of the dependence of the cost on the number of function calls is shown in Figure 13.

The following observations can be made:The first step of the algorithm requires 10 function calls because the function is 10 dimensional (five waypoints with two coordinates being optimized).The relationship between the initial cost and the final cost is not evident, but it does occur. It can be generalized that for most cases, the lower the initial cost, the lower the final cost.For call counts greater than 100, there is no significant further cost reduction in most cases.

A view of the mainlobe region for the worst initial situation, the worst final situation, and the best final situation is shown in Figure 14, Figure 15 and Figure 16, respectively. For comparison, the initial situation for the worst final situation is presented in Figure 17.

For the selected cases, a comparison between the initial and final trajectory was presented in Figure 18.

It can be observed that the best final case (#1) has the biggest change between the initial and final positions. Looking at Figure 13, it can be confirmed that this corresponds to a significant reduction in costs. However, in the worst final case (#8), the difference is almost unnoticeable, but the cost reduction is also significant (albeit much lower than in the case discussed above). This leads to the conclusion that relatively small changes in waypoint positions can lead to observable changes in cost. Investigating this case further by looking at Table 3 reveals that, in case #8, the optimization led to an improved vertical resolution at the price of a worsening Integrated Sidelobe Ratio (ISLR), which was due to a slight change in the PSF shape which caused some portion of the energy originally classified as the mainlobe to be classified as a sidelobe at some point during the optimization (note the abrupt change in cost around step 400 of the optimization—Figure 13). This can be confirmed by comparing Figure 15b with Figure 17b, which are almost identical.

In summary, the following conclusions are drawn from this observation:In some cases, a small change in the position of the points causes a noticeable change in the cost, but the actual change in the PSF is small and does not have a significant impact on the actual imaging parameters. Using at least a few initial values helps reduce the impact of such cases on the final result.The appropriate formulation of the weights of the components of the cost function is key to achieving the expected results.Minor inaccuracies in the navigation of the radar platform or external factors that alter the trajectory, such as wind, do not significantly affect the final quality of the image. It should be emphasized that this conclusion applies to the inaccuracy of platform guidance, not to the inaccuracy of platform position determination. The inaccuracy of the platform position determination results in a blurring of the imaging proportional to the position determination error and inversely proportional to the carrier wavelength [6].

### 3.2. Comparison with Reference

In this experiment, the best of the previously determined trajectories was compared with the MBSAR trajectories for two, three, four, and five baselines, demonstrated in Figure 19, Figure 20, Figure 21 and Figure 22, respectively. The optimized trajectory is shown in Figure 23. Compared to the previous section, the PSF is shown in a wider range so that the sidelobes, whose shapes differ significantly between the non-rectilinear and MBSAR trajectories, can be clearly seen.

Table 4 compares the PSF quality parameters for the previously mentioned scenarios. It should be noted that the time was determined only for the visible portions of the trajectory, which means that if the visible portion of the MBSAR trajectory was not treated as a slice of the CSAR trajectory, the flight time would be longer. The cost was determined according to the same criteria for each trajectory. The difference in cost for trajectory #1 compared to Table 3 is due to the wider range of the PSF determination which translates into a larger Integrated Sidelobe Ratio (ISLR) value.

By analyzing the above results, it can be seen that

The 2-pass and 3-pass trajectories have close and strong sidelobes (grating lobes), making it impossible to obtain three-dimensional (3D) imaging without additional processing, such as phase unwrapping.In terms of the flight time, the optimized trajectory ranks between the 3-pass and 4-pass trajectories.The optimized trajectory is characterized by the lowest Peak Sidelobe Ratio (PSLR) (comparable to the 2-pass).The optimized trajectory is characterized by the highest Integrated Sidelobe Ratio (ISLR) (comparable to the 5-pass).

Based on the results presented, it can be concluded that the presented method can be applied in scenarios where the benefits of a shorter flight time outweigh the disadvantages of a higher Integrated Sidelobe Ratio (ISLR). It should be noted that the PSF and quality parameter values also depend on the radar parameters and the geometry of the specific scenario.

## 4. Discussion

This paper presents an algorithm to determine the optimized trajectories for drone-based three-dimensional (3D) Synthetic Aperture Radar (SAR). The simulation results are presented, which confirm that trajectories with unevenly spaced waypoints allow shorter flight times than the traditionally used CSAR/MBSAR [2] and a lower Peak Sidelobe Ratio (PSLR) than the sinusoidal/hat maneuver trajectory [12].

On the basis of the results presented, it can be concluded that in order to determine a low-cost trajectory, many calls to the function that determines the PSF for a given trajectory are needed. This opens a field for further research, allowing to determine the tradeoff between the number of function calls for a given starting point and the number of different starting points.

It should be noted that even the high computational complexity of the presented algorithm is not its significant drawback from a practical point of view, because the trajectory of the radar platform can be determined long before the measurement campaign, at the stage of its planning.

The algorithm presented can be placed in a broader context: on the one hand, to use it in the trajectory planning work of a radar platform [20] to add three-dimensional (3D) imaging capability, and, on the other hand, to use the existing knowledge of three-dimensional (3D) Synthetic Aperture Radar (SAR) imaging using sparse apertures to improve the quality of acquired images [2,21,22,23]. Taking [20] into account, it may also be desirable to further develop the proposed algorithm, to be used by drone swarms [24].

## Figures and Tables

**Figure 1 sensors-22-06990-f001:**
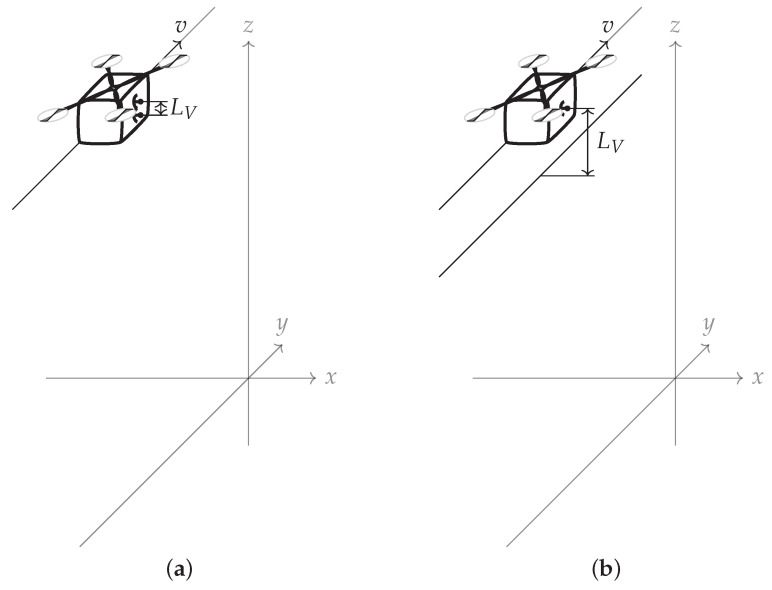
Traditional Interferometric Synthetic Aperture Radar (InSAR) geometries: (**a**) Dual antenna, single pass. (**b**) Single antenna, dual pass. *v* denotes the direction of motion of the radar platform, $L_{V}$ is the baseline, and *x*, *y*, *z* are Cartesian coordinates.

**Figure 2 sensors-22-06990-f002:**
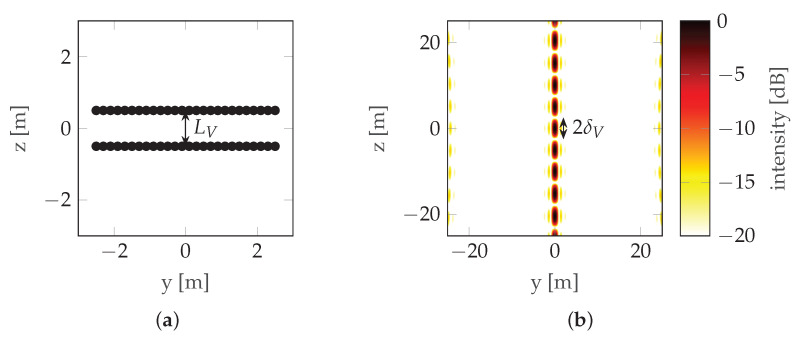
InSAR example: (**a**) Trajectory. (**b**) PSF. $L_{V}$ is the baseline and $\delta_{V}$ is the vertical resolution.

**Figure 3 sensors-22-06990-f003:**
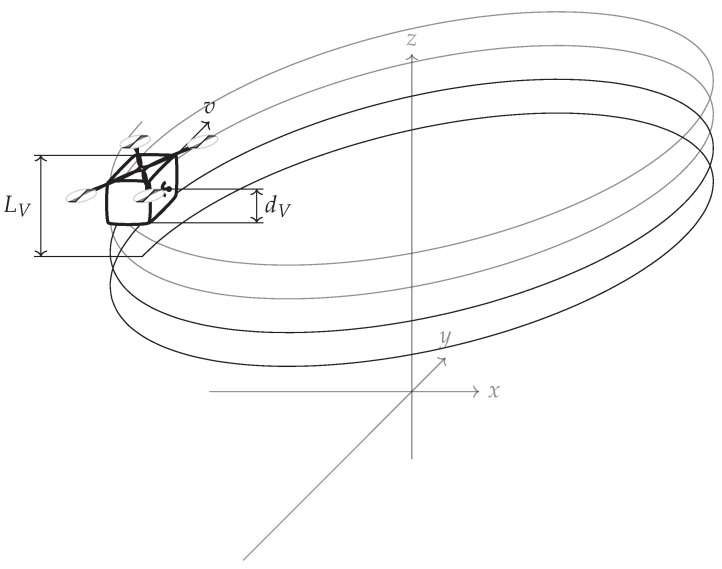
Circular SAR geometry. *v* denotes the direction of motion of the radar platform, $L_{V}$ is the vertical synthetic aperture size (the distance between the highest and lowest pass), and $d_{V}$ is the vertical sampling distance (the distance between consecutive passes).

**Figure 4 sensors-22-06990-f004:**
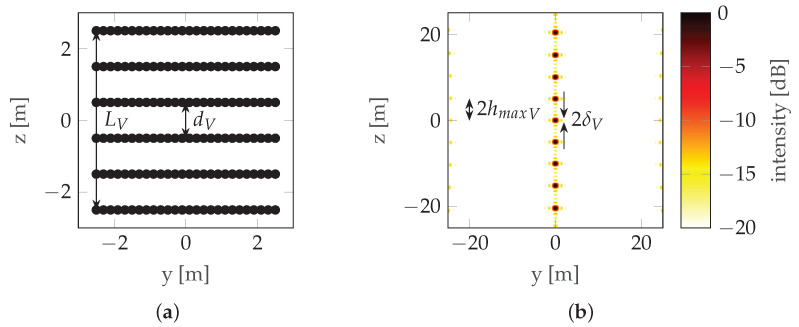
CSAR/MBSAR example: (**a**) Trajectory. (**b**) PSF. $L_{V}$ is the vertical synthetic aperture size (the distance between the highest and lowest pass) and $d_{V}$ is the vertical sampling distance (the distance between consecutive passes).

**Figure 5 sensors-22-06990-f005:**
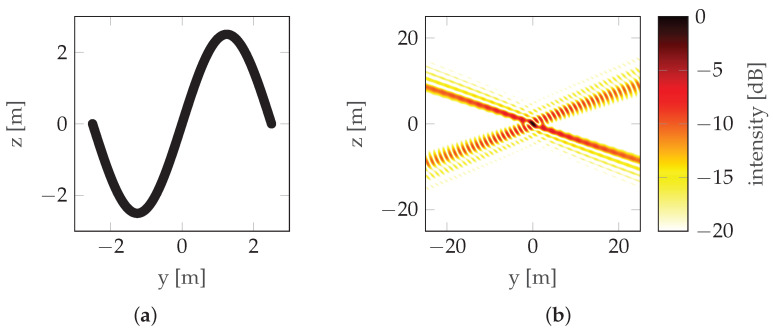
Sinusoidal trajectory/HAT maneuver example: (**a**) Trajectory. (**b**) PSF.

**Figure 6 sensors-22-06990-f006:**
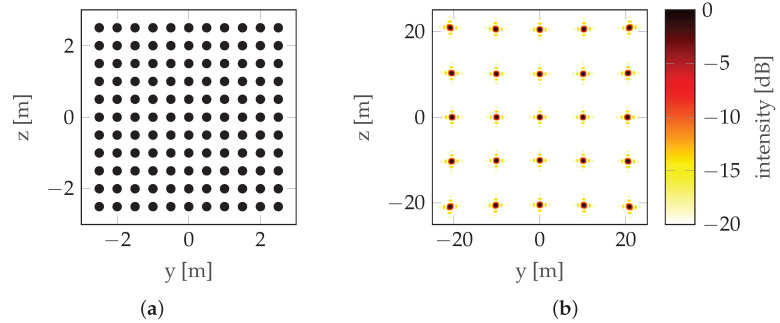
Evenly sampled aperture example: (**a**) Sampling points (“Trajectory”). (**b**) PSF.

**Figure 7 sensors-22-06990-f007:**
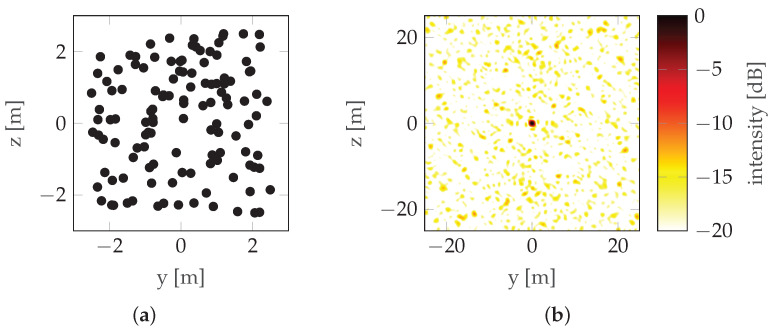
Randomly sampled aperture example: (**a**) Sampling points (“Trajectory”). (**b**) PSF.

**Figure 8 sensors-22-06990-f008:**
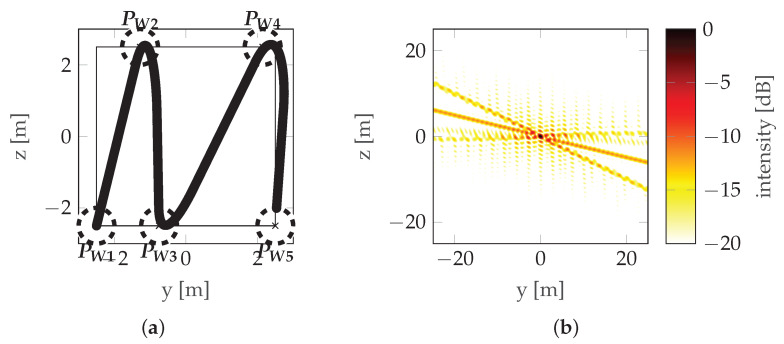
Real trajectory example: (**a**) Trajectory with waypoints denoted by $P_{W}$ and the acceptance circles marked with a dotted line. (**b**) PSF.

**Figure 9 sensors-22-06990-f009:**
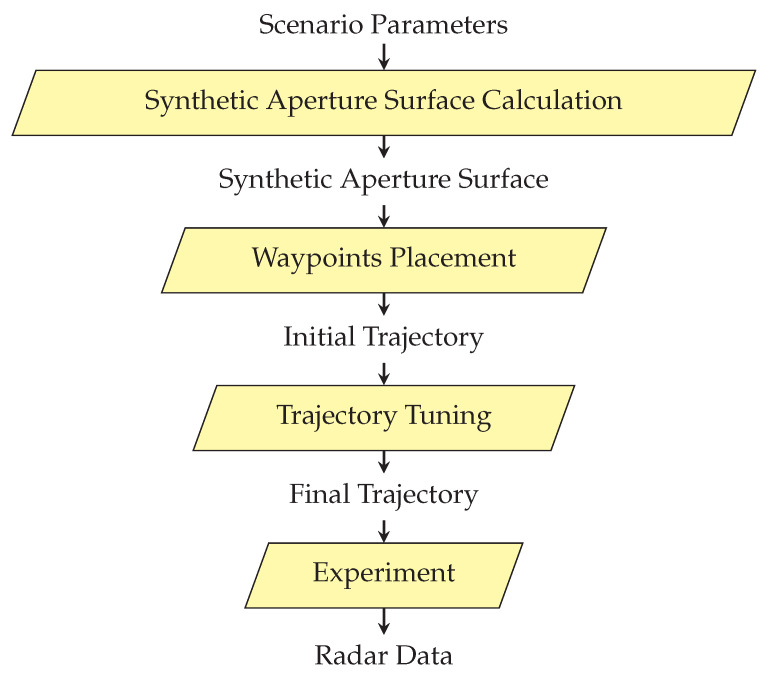
General diagram of three-dimensional (3D) Synthetic Aperture Radar (SAR).

**Figure 10 sensors-22-06990-f010:**
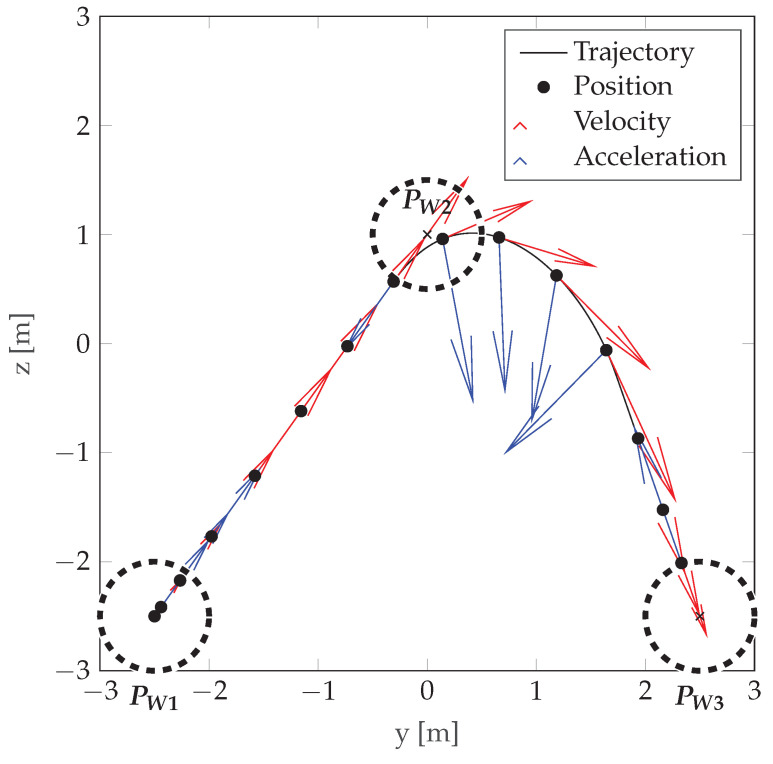
An example of the motion model used with acceleration and velocity shown in the figure. Waypoints are denoted by $P_{W\cdot }$ and the acceptance circles are marked with a dotted line.

**Figure 11 sensors-22-06990-f011:**
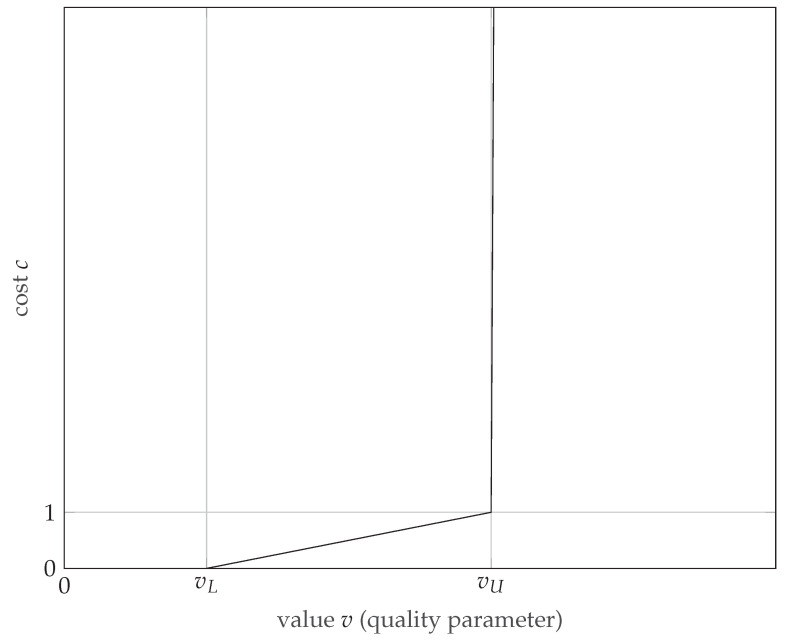
Range-type transfer function used to determine the cost related to the quality parameters.

**Figure 12 sensors-22-06990-f012:**
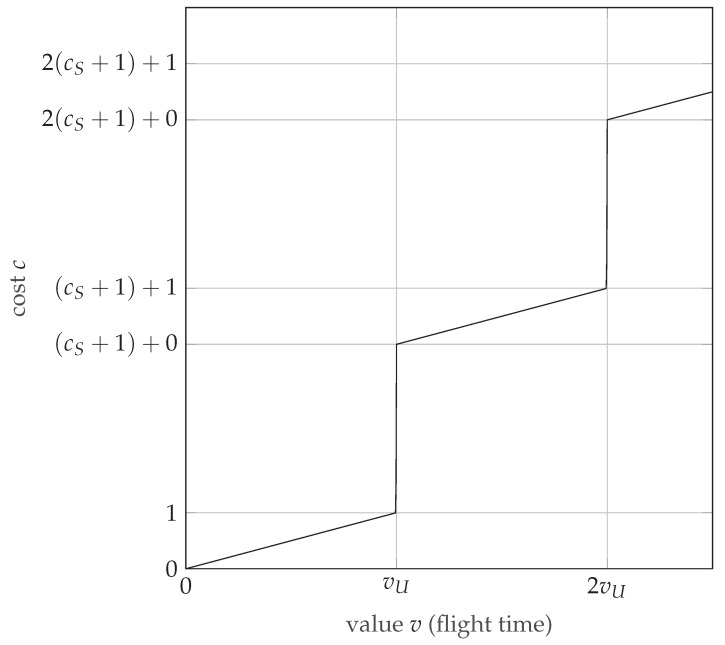
Step-type transfer function used to determine the cost related to flight time.

**Figure 13 sensors-22-06990-f013:**
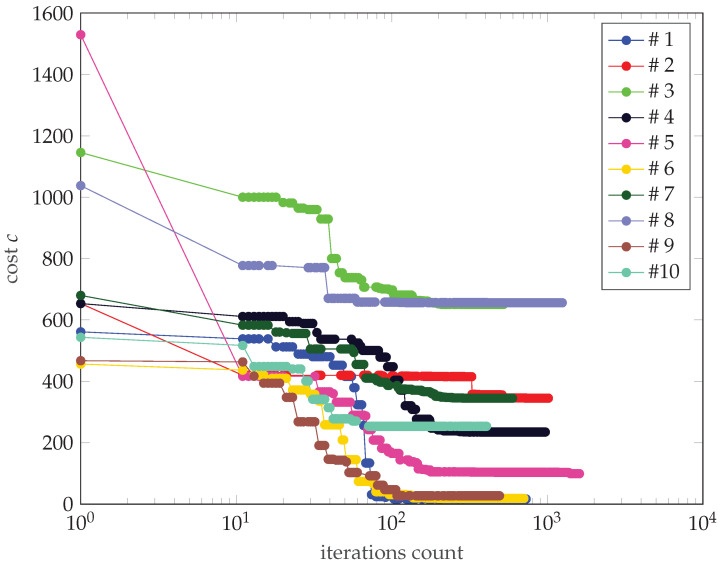
Cost versus function calls. Different starting points are denoted with different colors.

**Figure 14 sensors-22-06990-f014:**
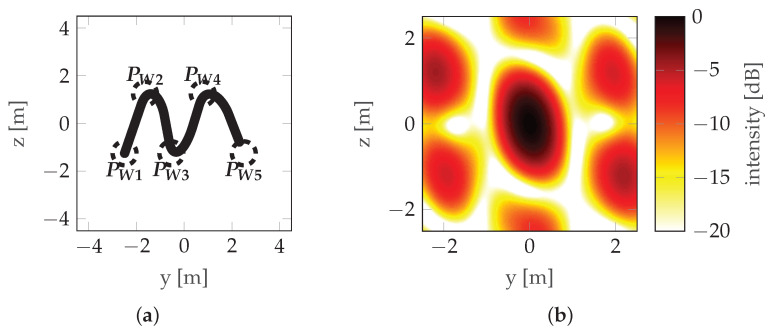
Worst initial situation—run #5: (**a**) Trajectory with waypoints denoted by $P_{W}$ and the acceptance circles marked with a dotted line. (**b**) PSF.

**Figure 15 sensors-22-06990-f015:**
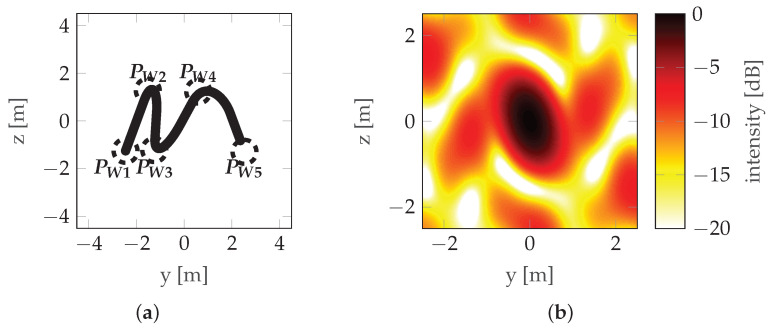
Worst final situation—run #8: (**a**) Trajectory with waypoints denoted by $P_{W}$ and the acceptance circles marked with a dotted line. (**b**) PSF.

**Figure 16 sensors-22-06990-f016:**
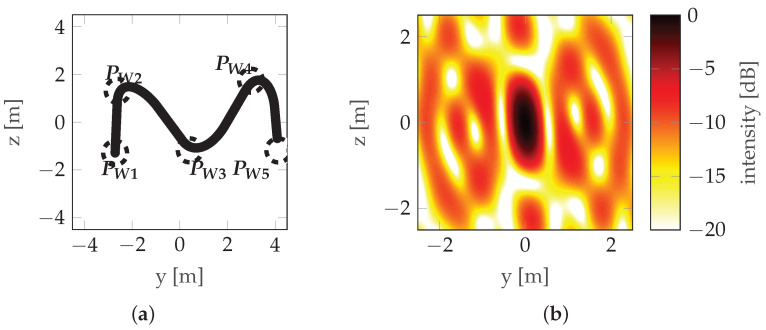
Best final situation—run #1: (**a**) Trajectory with waypoints denoted by $P_{W}$ and acceptance circles marked with a dotted line. (**b**) PSF.

**Figure 17 sensors-22-06990-f017:**
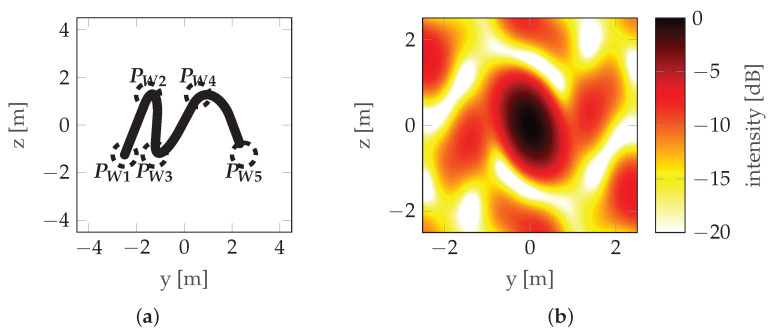
Initial state for the worst final situation—run #8: (**a**) Trajectory with waypoints denoted by $P_{W}$ and acceptance circles marked with a dotted line. (**b**) PSF.

**Figure 18 sensors-22-06990-f018:**
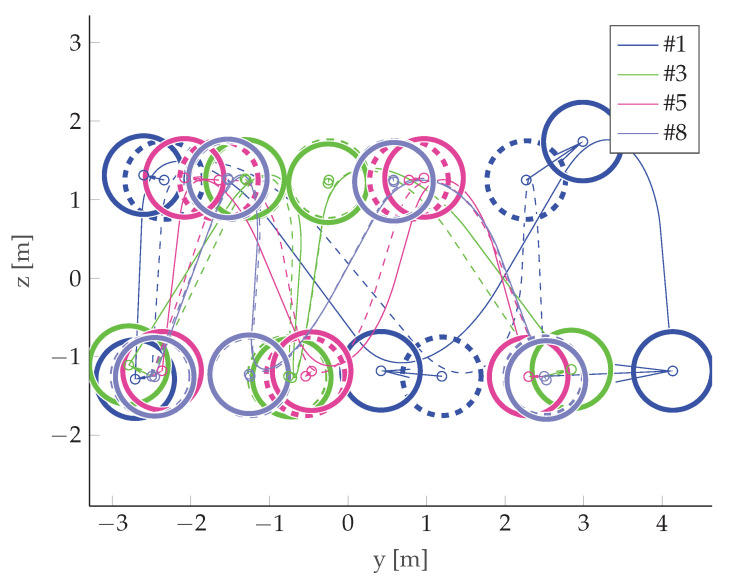
Comparison between the initial and final trajectory for selected cases. The dashed lines represent the initial trajectory, the solid lines the final trajectory, and the dashed and solid circles represent the initial and final acceptance circles, respectively.

**Figure 19 sensors-22-06990-f019:**
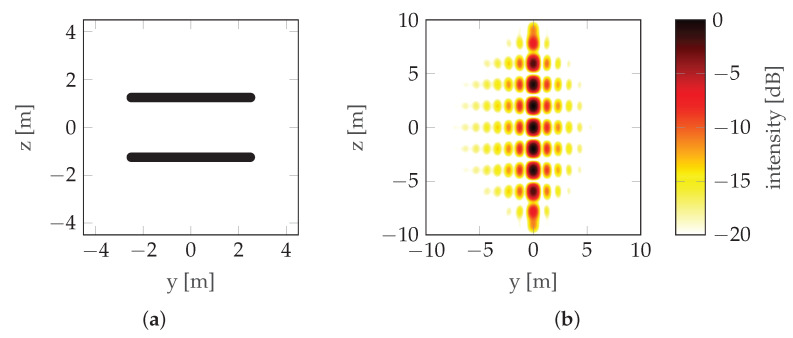
Reference scenario with two passes: (**a**) Trajectory. (**b**) PSF.

**Figure 20 sensors-22-06990-f020:**
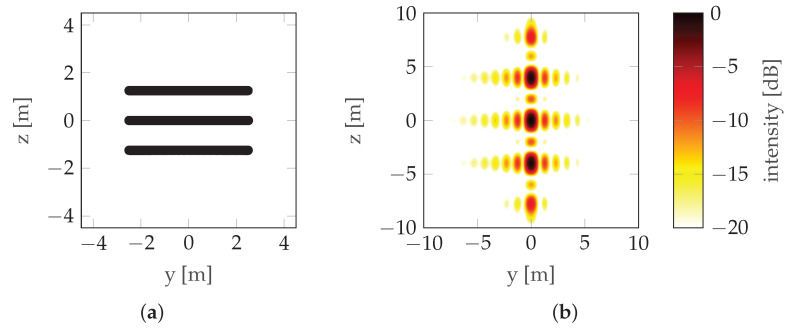
Reference scenario with three passes: (**a**) Trajectory. (**b**) PSF.

**Figure 21 sensors-22-06990-f021:**
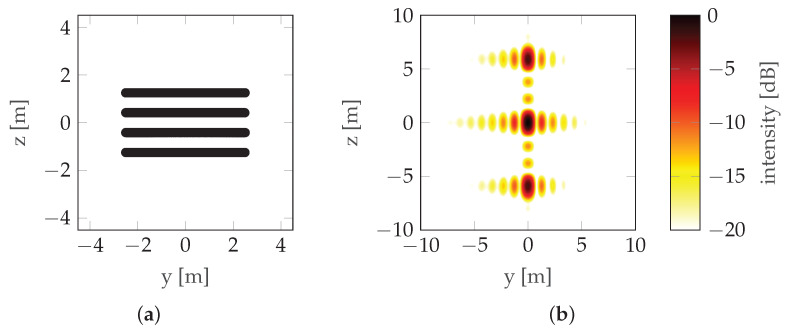
Reference scenario with four passes: (**a**) Trajectory. (**b**) PSF.

**Figure 22 sensors-22-06990-f022:**
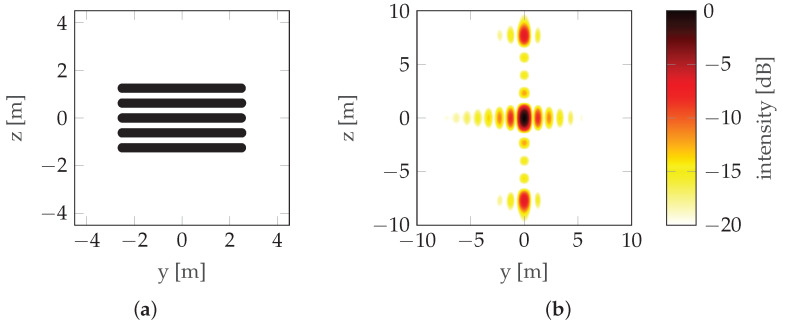
Reference scenario with five passes: (**a**) Trajectory. (**b**) PSF.

**Figure 23 sensors-22-06990-f023:**
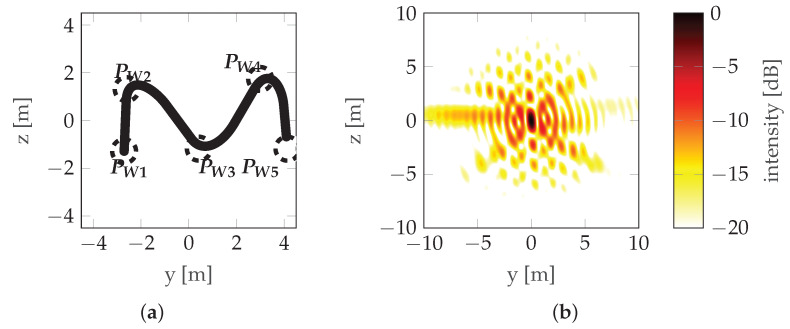
Best final situation—run #1—wide PSF range: (**a**) Trajectory with waypoints denoted by $P_{W}$ and the acceptance circles marked with a dotted line. (**b**) PSF.

**Table 1 sensors-22-06990-t001:** Test scenario and radar parameters.

Parameter	Value
ROI horizontal dimension $D_{H}$	10 m
ROI vertical dimension $D_{V}$	10 m
distance to ROI $r_{0}$	100 m
max. upward acceleration $a_{↑\max}$	3 m/s^2^
max. downward acceleration $a_{↓\max}$	8 m/s^2^
max. vertical acceleration $a_{↔\max}$	5 m/s^2^
max. upward velocity $v_{↑\max}$	4 m/s
max. downward velocity $v_{↓\max}$	8 m/s
max. vertical velocity $v_{↔\max}$	6 m/s
carrier frequency fc	3 GHz
bandwidth *B*	300 MHz
horizontal beamwidth $\theta_{H}$	10°
vertical beamwidth $\theta_{V}$	20°
PRF	1 kHz

**Table 2 sensors-22-06990-t002:** Optimization parameters.

Parameter	Lower Value	Upper Value	Weight
${\delta_{H}}_{L/U}$	$0.9\;\delta_{H}$	$1.1\;\delta_{H}$	0.3
${\delta_{V}}_{L/U}$	$0.9\;\delta_{V}$	$1.1\;\delta_{V}$	0.3
PSLR	$0.9\;PSLR_{ref}$	$1.1\;PSLR_{ref}$	0.2
ISLR	$ISLR_{ref}$	$ISLR_{ini}$	0.1
*t*	-	$t_{ref}$	0.1

**Table 3 sensors-22-06990-t003:** Trajectory generation result parameters for ten different start points.

#	$\delta_{H}$		$\delta_{V}$		PSLR		ISLR		cost *c*	
	ini	fin	ini	fin	ini	fin	ini	fin	ini	fin
1	0.94	0.82	1.97	1.93	0.71	0.46	20.37	22.90	561.28	16.63
2	1.32	1.12	2.02	1.70	0.60	0.60	14.21	15.80	653.68	345.26
3	1.79	1.49	1.66	1.60	0.52	0.48	9.64	11.89	1145.60	650.58
4	1.24	1.21	1.85	1.73	0.66	0.50	14.12	14.15	653.42	234.86
5	1.19	1.14	3.68	2.09	0.59	0.47	12.08	13.68	1529.34	99.23
6	0.99	1.08	1.82	1.49	0.66	0.47	16.09	17.28	456.42	18.50
7	1.24	1.20	1.80	1.87	0.67	0.55	12.03	11.98	679.96	345.06
8	1.59	1.53	2.58	1.86	0.47	0.45	9.53	10.36	1037.71	656.06
9	1.25	1.12	1.69	2.11	0.57	0.43	13.64	12.68	467.29	27.27
10	1.39	1.27	1.68	1.87	0.51	0.45	11.85	11.59	543.47	253.53

**Table 4 sensors-22-06990-t004:** Quality parameter comparison for MBSAR trajectories and an optimized trajectory.

Scenario	$\delta_{H}$	$\delta_{V}$	PSLR	ISLR	Time *t*	Cost *c*
MBSAR 2-pass	0.83	1.02	0.99	23.93	3.05	1258.84
MBSAR 3-pass	0.83	1.35	0.95	13.62	4.58	1131.92
MBSAR 4-pass	0.89	1.52	0.79	9.66	6.10	766.22
MBSAR 5-pass	0.87	1.62	0.48	7.55	7.63	37.84
optimized #1	0.57	1.81	0.46	25.95	5.22	35.94

## Data Availability

Not applicable.

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
