# Peer review of "Drone-Based 3D Synthetic Aperture Radar Imaging with Trajectory Optimization"

_sensors, 2022, doi:10.3390/s22186990_

Round 1

Reviewer 1 Report

This paper presents a trajectory determination and optimization method for multirotor equipped with a single-channel radar to obtain 3D Synthetic Aperture Radar imaging. After carefully read this paper, I cannot recommend this work to be accepted before the following concerns being addressed:

1.     In Section 1, the discussion on the related work is insufficient. It is suggested to add a more detailed introduction on 3D imaging algorithms.

2.     In Section 1.1, please give more rigorous technical details on the SAR signal model.

3.     In fig.4, you mentioned “It can be seen that increasing the horizontal synthetic aperture length (LH) by adding more baselines increases the resolution without increasing the height ambiguity.” How to draw a such conclusion? I don’t find any comparison.

4.     How to optimize the compound cost function? Please give algorithmic steps.

5.     Please give a clear description on fig.11 and 12.

6.     In equation (40), I suggest giving a clear definition of Pw.

7.     The experiments are too simple, it is suggested to compare with the existing SOTA methods, or even conventional ones, so that the effectiveness of your methodology can be convictive.

8.     More testing conditions should be taken into consideration, for instance, what happens with motion errors?

9.     It will be more convictive if the proposed method can be validated on measured experiments.

Author Response

Dear Reviewers,

As authors, we would like to express our great appreciation for your work. Your invaluable comments not only helped us to correct the mistakes that we were unable to avoid when submitting the first version of the article, but most importantly, your expert suggestions allowed us to better direct our research, and consequently made the revised version of the article better than it could have been without your input.

We have carefully read each of your comments and have addressed each below. All of your comments have been incorporated directly, as you suggested. We hope that we have solved all indicated problems.

We regret that we could not take into account Reviewer 1's comment suggesting that we perform real-life experiments. This is a very pertinent comment, and certainly, the results of the experiments would have been valuable for this research work. However, the current stage of this research is at a low technology readiness level (TRL), and this validation was out of scope at this stage. We plan to do this in future, based on the theory presented in this paper. However, it will take significant additional effort, and we hope that we will be able to share such results in our next research papers. We believe that the results presented in this paper will also enable other scientific teams to use them in their work, including experiments.

Sincerely

Jedrzej Drozdowicz and Piotr Samczynski

Reviewer 1 [C]omments and author [A]nswers:

R1C1. In Section 1, the discussion on the related work is insufficient. It is suggested to add a more detailed introduction on 3D imaging algorithms.

R1A1. Thank you for this comment, it was important to stress that traditional matched filter approach can’t be used for non-rectilinear trajectories. Subsection 1.3. “Imaging algorithms” with an introduction to 3D imaging algorithms was added.

R1C2. In Section 1.1, please give more rigorous technical details on the SAR signal model.

R1A2. This research is not tied to any specific signal model, such as FMCW. However this was added for reference in subsection 1.1. “Synthetic Aperture Radar Signal model” and explained.

R1C3. In fig.4, you mentioned “It can be seen that increasing the horizontal synthetic aperture length (LH) by adding more baselines increases the resolution without increasing the height ambiguity.” How to draw a such conclusion? I don’t find any comparison.

R1A3. We added the explanation with references to the figures and equations and fixed the typo that mixed “Horizontal” with “Vertical”. The resubmitted version should be more clear.

R1C4. How to optimize the compound cost function? Please give algorithmic steps.

R1A4. Thank you for this comment, this is an important part of this work. Algorithmic steps were provided in subsection 2.10. “Optimization”.

R1C5. Please give a clear description on fig.11 and 12.

R1A5. We extended the figure caption and labeled axes in a proper fashion. See also R2C5.

R1C6. In equation (40), I suggest giving a clear definition of Pw.

R1A6. The equation was fixed and description was provided.

R1C7. The experiments are too simple, it is suggested to compare with the existing SOTA methods, or even conventional ones, so that the effectiveness of your methodology can be convictive.

R1A7. Thank you for this valuable comment. Indeed a comparison with SOTA is needed to justify the research on this method. We performed additional simulation and presented a comparison with typical MBSAR/CSAR trajectory. We have added subsection 3.2. “Comparison with Reference” where we commented on the results.

R1C8. More testing conditions should be taken into consideration, for instance, what happens with motion errors?

R1A8. This indeed would be interesting, however on one hand this opens a very broad scope for research, and on the other this topic is mostly covered by other literature. For this method motion measurement errors (the inability to precisely determine the position of the radar) will degrade the image quality just as in the case of any other SAR imaging method. Steering errors are however important and a paragraph was added on that at the end of “Results”, along with a figure showing comparison between initial and final trajectories (Figure 18 in revised version) and a figures showing corresponding PSFs (Figures 15 and 16 in revised version). We have added a comment on the impact of trajectory change on the PSF on the example of case #5.

R1C9. It will be more convictive if the proposed method can be validated on measured experiments.

R1A9. Yes, of course. This is however hard to accomplish given the limited access to the appropriate equipment by the authors. We sincerely hope that the description of the method in this paper is detailed enough so that other research teams with more experience with drone-borne SAR deployment can build on this research and that even without experimental verification it has substantial value to the scientific community.

Reviewer 2 Report

The article describes the Drone-Based 3D Synthetic Aperture Radar Imaging with Trajectory Optimization. Although the authors have explained all the aspects related to the trajectory optimization, still some minor changes are required, as mentioned below;

1. The manuscript needs moderate grammar and spell check. 

2. Line 11: Avoid the start of the introduction with words like "Through". Modify the sentence accordingly.  

3. In the introduction part of abbreviations "SAR" and "UAV" are used. Describe the full forms at the beginning. 

4. In the manuscript, most of the notations are in various equations used without their explanations. Check it properly and explain it. 

5. Figure 11: Just notations are mentioned, such as c and v. Explain their meaning at the figure axis. check for other figures also. 

Author Response

Dear Reviewers,

As authors, we would like to express our great appreciation for your work. Your invaluable comments not only helped us to correct the mistakes that we were unable to avoid when submitting the first version of the article, but most importantly, your expert suggestions allowed us to better direct our research, and consequently made the revised version of the article better than it could have been without your input.

We have carefully read each of your comments and have addressed each below. All of your comments have been incorporated directly, as you suggested. We hope that we have solved all indicated problems.

We regret that we could not take into account Reviewer 1's comment suggesting that we perform real-life experiments. This is a very pertinent comment, and certainly, the results of the experiments would have been valuable for this research work. However, the current stage of this research is at a low technology readiness level (TRL), and this validation was out of scope at this stage. We plan to do this in future, based on the theory presented in this paper. However, it will take significant additional effort, and we hope that we will be able to share such results in our next research papers. We believe that the results presented in this paper will also enable other scientific teams to use them in their work, including experiments.

Sincerely

Jedrzej Drozdowicz and Piotr Samczynski

Reviewer 2 [C]omments and author [A]nswers:

R2C1. The manuscript needs moderate grammar and spell check. 

R2A1. The manuscript underwent additional grammar and spell check before resubmission.

R2C2. Line 11: Avoid the start of the introduction with words like "Through". Modify the sentence accordingly.

R2A2. The sentence was modified, thank you for this remark.

R2C3. In the introduction part of abbreviations "SAR" and "UAV" are used. Describe the full forms at the beginning.

R2A3. We are sorry for overseeing that. Latex-glossaries package was used in resubmitted paper for all abbreviations to ensure they are expanded at first use. It might have added some clutter to the diff.

R2C4. In the manuscript, most of the notations are in various equations used without their explanations. Check it properly and explain it.

R2A4. All the equations were carefully checked for unexplained notations before resubmission. Missing explanations were added.

R2C5. Figure 11: Just notations are mentioned, such as c and v. Explain their meaning at the figure axis. check for other figures also.

R2A5. Thank you for spotting this. This was changed for Figures 11, 12 and 13. Moreover „db” was changed to „intensity [dB]” on every Figure that presented PSF. Only „x [m]”, „y [m]” and „z [m]” were left unexplained on every picture as their meaning is obvious and their are explained at the beginning. Moreover symbols used in the Figures with geometries were explained in the figure caption for clarity.

Reviewer 3 Report

Proposed paper addresses an open topic for a-priori optimization of SAR imaging. Methodology is interesting and conclusions are good. In my opinion there's a lack of clarity in quality metrics definition: all my observations rely on this aspect. Once such paragraph is rewritten, I will suggest this paper for publication.

Idea is good, but to well appreciate the paper I need more details about some algorithmical steps.

Issues:

par.2.4 needs a better clarification, as it is in my opinion the main point of the entire paper. I suppose deltaV and deltaH are the resolutions achieved with a given set of waypoints, but this isnt well depicted throughout text. (30)-(31) are defined respect the average value of P0Vz and P0Hy? Where is dependancy towards alpha? What is alpha? Note that alpha is never defined before 173.

I think it is mandatory to add a paragraph explaining better the steps between (29) and (34).

For (34) pixel adjacency is expressed by a distance of "1". Thinking at image distinguishability and effect of melting of different mainlobes, maybe fixed "1" is too much optimistic? If it is possible, argument this.

211: vL to vU"X" "X" is typo?

(38) maybe it is better to define range-cost function respect range "r" domain

(39) maybe it is better to define function respect time "t" domain

It could be possible to describe better which variation function is applied to each point coordinate towards successive iterations counts? It is interesting that the "delta" between initial cost and final cost is similar between all different starting sets. Can this be due to the limitation of domain spanning for the variation of each waypoint?

I think it would be very interesting and helpful to provide a graphical representation of the 10 different sets of starting waypoints. Furthermore, it would be interesting to put graphically in comparison the starting positions of waypoints with the final optimized ones, at least for a couple of waypoints sets.

Author Response

Dear Reviewers,

As authors, we would like to express our great appreciation for your work. Your invaluable comments not only helped us to correct the mistakes that we were unable to avoid when submitting the first version of the article, but most importantly, your expert suggestions allowed us to better direct our research, and consequently made the revised version of the article better than it could have been without your input.

We have carefully read each of your comments and have addressed each below. All of your comments have been incorporated directly, as you suggested. We hope that we have solved all indicated problems.

We regret that we could not take into account Reviewer 1's comment suggesting that we perform real-life experiments. This is a very pertinent comment, and certainly, the results of the experiments would have been valuable for this research work. However, the current stage of this research is at a low technology readiness level (TRL), and this validation was out of scope at this stage. We plan to do this in future, based on the theory presented in this paper. However, it will take significant additional effort, and we hope that we will be able to share such results in our next research papers. We believe that the results presented in this paper will also enable other scientific teams to use them in their work, including experiments.

Sincerely

Jedrzej Drozdowicz and Piotr Samczynski

Reviewer 3 [C]omments and author [A]nswers:

R3C1. par.2.4 needs a better clarification, as it is in my opinion the main point of the entire paper. I suppose deltaV and deltaH are the resolutions achieved with a given set of waypoints, but this isnt well depicted throughout text. (30)-(31) are defined respect the average value of P0Vz and P0Hy? Where is dependancy towards alpha? What is alpha? Note that alpha is never defined before 173.

R3A1. Thank you for this comment. The description was clarified and expanded. See also R3A2.

R3C2. I think it is mandatory to add a paragraph explaining better the steps between (29) and (34).

R3A2. Thank you for this remark. Subsection 2.5. “Mainlobe extraction” was added and it improved the readability of this section.

R3C3. For (34) pixel adjacency is expressed by a distance of "1". Thinking at image distinguishability and effect of melting of different mainlobes, maybe fixed "1" is too much optimistic? If it is possible, argument this.

R3A3. The pixel adjacency is 1 because it can’t be smaller. Note that there is no need for sub-pixel resolution as physical pixel size is a simulation parameter and can be set to arbitrarily small value (which will increase computation time). As this was unclear, clarification was added.

R3C4. 211: vL to vU"X" "X" is typo?

R3A4. Yes, „X” was a typo. We are sorry for overlooking that.

R3C5. (38) maybe it is better to define range-cost function respect range "r" domain

R3A5. It is not a „range-cost” function, but a range type cost function, where „range” denotes the „range of values” between vL and vU. We couldn’t find a better word than „range”, but as it can be misleading we added „quality parameter” to the value in Equation description and on the graph. This function can be applied to various parameters, such as resolution, PSLR and ISLR.

R3C6. (39) maybe it is better to define function respect time "t" domain

R3A6. This is a good remark, however as the previous cost function can be applied to various parameters, such as resolution, PSLR and ISLR, we would keep using „value” for consistency. However we added „flight time” both to the graph and description.

R3C7. It could be possible to describe better which variation function is applied to each point coordinate towards successive iterations counts? It is interesting that the "delta" between initial cost and final cost is similar between all different starting sets. Can this be due to the limitation of domain spanning for the variation of each waypoint?

R3A7. An analysis was provided at the end subsection 3.1. “Performance Analysis”. Very interesting remark about the „delta”, it is in fact around 500 in most of the cases, but there also the cases where it is around 1500 (case 5) or 300 (case 10). We will investigate it further, but at this point it would be too speculative to draw any strong conclusions.

R3C8. I think it would be very interesting and helpful to provide a graphical representation of the 10 different sets of starting waypoints. Furthermore, it would be interesting to put graphically in comparison the starting positions of waypoints with the final optimized ones, at least for a couple of waypoints sets.

R3A8. Thank you, that is indeed interesting and helpful. We have provided a graphical comparison of initial and final waypoints and trajectories for 4 selected cases (unfortunately such a graph for 10 cases is unreadable, however a set of 4 provides a good insight). A comment was added to text. Additionally we changed color pattern in Figure 13 to be more distinctive and match match those of the new figure.

Round 2

Reviewer 1 Report

Thanks the authors for carefully revised and replied according to my comments, I have no new comment. The manuscript can be receipted in this form.

Reviewer 3 Report

The authors understood all my review points and implemented all improvements that I suggested. They did a very good job which totally improved the overall quality of paper.